# Deep Learning-Based Thermal Image Analysis for Pavement Defect Detection and Classification Considering Complex Pavement Conditions

Cheng Chen [1], Sindhu Chandra [2], Yufan Han [3] and Hyungjoon Seo [2,*]

1 Department of Civil Engineering, Xi'an Jiaotong Liverpool University, Suzhou 215123, China; Cheng.Chen19@student.xjtlu.edu.cn
2 Department of Civil Engineering and Industrial Design, University of Liverpool, Liverpool L69 3BX, UK; S.Chandra@liverpool.ac.uk
3 Department of Computer Science, Xi'an Jiaotong Liverpool University, Suzhou 215123, China; Yufan.Han20@student.xjtlu.edu.cn
* Correspondence: hyungjoon.seo@liverpool.ac.uk; Tel.: +44-151-795-7312

**Abstract:** Automatic damage detection using deep learning warrants an extensive data source that captures complex pavement conditions. This paper proposes a thermal-RGB fusion image-based pavement damage detection model, wherein the fused RGB-thermal image is formed through multi-source sensor information to achieve fast and accurate defect detection including complex pavement conditions. The proposed method uses pre-trained EfficientNet B4 as the backbone architecture and generates an argument dataset (containing non-uniform illumination, camera noise, and scales of thermal images too) to achieve high pavement damage detection accuracy. This paper tests separately the performance of different input data (RGB, thermal, MSX, and fused image) to test the influence of input data and network on the detection results. The results proved that the fused image's damage detection accuracy can be as high as 98.34% and by using the dataset after augmentation, the detection model deems to be more stable to achieve 98.35% precision, 98.34% recall, and 98.34% F1-score.

**Keywords:** pavement defect detection; machine learning; thermal analysis; multichannel image fusion

## 1. Introduction

The growth of urban traffic and the consequent increase in traffic volume over the years have made the timely maintenance of pavements extremely important. Repetitive traffic loads [1], rapid temperature changes [2], and reflection from base layers [3] are deemed to contribute directly to pavement damages. Also, water ingress into initial pavement cracks can deepen the damage resulting in distresses like potholes, even pavement structural failures [4]. Thus, timely maintenance can not only ensure safe operation but also increase and the service life of pavements. The current pavement crack detection is manual with subjective human interpretation and reparation mainly involve filling of the crack with sealant. Although, automated pavement detection systems have been studied for many years, previous researches were primarily focused on crack extraction. However, for the actual complex road conditions, the existing methods have limited error detection rates to identify all kinds of cases [5]. Multi-sensor fusion processing idea for complex road conditions was considered where acceleration sensors [6], infrared sensors [7], multi-vision cameras [8], and 3D laser scanning [9] can provide additional identification information to the optical images of the pavement.

Automated pavement detection has undergone several significant technological changes, and digital image-based methods have been widely used for pavement crack detection and segmentation. The difference in grayscale values of crack pixels and background of digital images makes segmentation as well as detection logical [10]. Other factors such as lighting

conditions, asphalt oil markings, and pavement markings, make pavement crack classification and segmentation are challenging as well. Thus, the use of image processing has the advantages of fast processing speed, low cost and high robustness compared to the use of laser scanning or 3D point clouds from multi-vision cameras. At the same time, due to the acquisition of images only, it may not be possible to achieve complete recognition under complex pavement scenes. Additionally, image recognition achieved through image processing and machine learning, wherein machine learning can be subdivided into traditional machine learning and deep learning [11]. These image processing methods do not require a model training process and usually involve the use of filters, morphological analysis, and statistical methods to detect cracks. These methods require careful filtering for the usage scenario and may not be robust to noise such as lighting, oil stains, etc. Zou et al. [12] proposed a crack tree noting the effect of lighting on pavement detection and proposed a shadow removal algorithm before crack extraction to eliminate the effect of shadows on the extraction results. However, the crack recognition requires the assistance of some machine learning algorithms such as SVM (Support Vector Machine), RBF (Radial Basis Function), KNN (K-Nearest Neighbor) and Random Decision Forest [10,13,14]. Also, statistical features, gray-level features, texture and shape features of cracked images are increasingly used for feature extraction of the images. In addition to considering block-level information, multi-scale information also gets used for multi-scale fusion crack detection (MFCD) to detect cracks [13]. Furthermore, principal component analysis (PCA)uses simplified number of feature levels to speed up the identification of a single crack block [10].

In recent years, artificial intelligence (AI) has become increasingly popular in automatic image processing and recognition, and deep learning has gained traction in object detection and segmentation [15]. Unlike basic machine learning, deep learning uses designed convolution to replace feature extraction in traditional machine learning with the powerful parallel computing capability of the graphics processing unit (GPU). Deep learning for recognition detection problems greatly improves the detection efficiency and the accuracy of recognition. Deep structures with many hidden layers, such as deep convolutional neural networks (DCNN), provide increased levels of feature abstraction to reflect the complexity of the data, and diverse convolutional layers of feature extraction increase the confidence of the features [16]. Deep learning also uses raw data for processing, end-to-end networks with minimal human intervention and prior assumptions, which increases the possibility of recognition detection for complex scenarios. Also, the image recognition network was transferred to the domain of pavement crack detection through transfer learning. Gopalakrishnan et al. [17] completed fully automated detection of pavement distress using the VGG16 network. And Cha et al. [18] proposed a DCNN architecture for detecting concrete cracks in intensity images under uneven illumination conditions. Similarly, Cha et al. [19] modified the original fast R-CNN architecture to detect and classify five defective concrete cracks. And Zhang et al. [20] transferred a network from a pre-trained AlexNet to facilitate the learning process to classify image patches as cracks, sealed cracks, and background regions. These deep learning models by transfer learning seek new application scenarios from established image recognition models, and their results show that the deep learning approach outperforms traditional image processing. However, these methods only focus on the recognition of the cracks themselves and do not identify and detect the complex factors such as oil markings, joints, manholes, etc. Majidifard et al. [21] combined YOLO (you only look once) deep learning framework and U-net as a two-step network to identify and extract cracks from the pavement image and made a detailed subdivision into eight types of cracks and potholes for the complex scenes of the pavement. Also, Chen et al. [6] also used a fusion network with acceleration using wavelet transform and VGG16 to identify and detect transverse cracks and manholes.

The pavement crack detection and even quantitative crack measurement in complex scenes have been further investigated by multi-sensors and deep learning. Zhou and Song [22] used DCNN and laser-scanned range images to identify cracks with depth mapping information to avoid the effect of oil stains and shadows on the pavement.

Also, Guan et al. (2021) used automatic pavement detection based on stereo vision and deep learning, and the 3D image dataset enabled effective identification of cracks and potholes, and the 3D image with depth information enabled volume measurement of potholes. A thermal image was also used for pavement crack detection, and the surface temperature distribution pattern is directly related to the pavement crack profile, which can be used as an indicator of crack depth [23]. Seo et al. [24] conducted an experimental study on the behavior of cracks by applying infrared thermal images, depending on the different widths of the columns, and confirmed that infrared thermal imagers can detect cracks [25]. Compared to multi-visual cameras or laser-scanned points or images, thermal imagers tend to have better real-time efficiency, low-cost characteristics, and the ability to directly process raw data into deep learning networks. Thermal imaging is certainly one of the auxiliary devices for practical pavement inspection. Georgia Tech Sensing Vehicle (GTSV) [26] was used to collect 3D pavement surface images to validate existing pavement detection methods. Mainstream models such as the Full Convolutional Network (FCN) model [27], U-Net [28], DeepCrack [29] with VGG16 backbone, and Pix2Pix [30] based on generative adversarial networks (GAN) were used for qualitative and objective evaluation of crack detection algorithms. Ali et al. [31] proposed a local weighting factor with sensitivity map to eliminate network bias and accurately predict sensitive pixels, a deep full convolutional neural network with better crack segmentation performance than U-Net was implemented. Fan et.al [32] proposed a medium hierarchical feature learning and inflated convolutional encoder-decoder architecture for crack detection with an end-to-end approach. Wang et.al [33] proposed a semantic segmentation framework for cracked images based on semi-supervised learning, which greatly reduces the workload of data annotation, and his proposed network can extract and merge information from multiple feature layers to improve the performance of the algorithm.

Based on previous crack detection studies, this paper analyses the statistical distribution characteristics of cracks in the optical and thermal images. This paper also proposes two improvements for pavement detection assisted by the thermal image: data augmentation and the detection system with fusion pre-processing. It may not be enough to support sufficient recognition accuracy of the complex scenes by a small dataset due to the absence of any existing infrared camera-assisted pavement detection database. Therefore, this paper uses a data augmentation method to augment the training set in the collected dataset to verify that augmented data can effectively increase detection accuracy. When using optical sensors for image-based pavement damage detection, the accuracy of the detection system was affected by the uneven light, shadow and oil marks. Point clouds are not available as real-time detection due to their full acquisition and long processing time. This paper also proposes a pavement crack detection algorithm based on EfficientNet [34], which gets tested using optical images, thermal images, and pre-processed fusion images to investigate the effectiveness of infrared cameras for recognizing complex scenes on a pavement in a side-by-side comparison.

## 2. Methodology and Materials

This section describes the methods and materials used in this study. This encompasses data collection, the proposed method and the specific steps of implementation including incorporation of specific technical tools.

### 2.1. Overall Procedure

Figure 1 shows the workflow of the proposed work. Each pavement damage was captured as an RGB image and thermal image (registered by the device at the acquisition stage). A 50% transparent thermal image was then superimposed on the RGB image to form a fused image, thus forming three distinct data sets for comparison experiments. Also, nine different categories of pavement damages were considered, and 500 images were captured per category (The diversity of the dataset ensures the robustness of the model in deep learning [35]. The image data was then put through data augmentation to increase the

data sample size. The data set was divided in the ratio of 6:2:2 to form a training set, test set, and validation set. The EfficientNet-based learning mechanism was then used to train the model using a ten-fold cross-validation training method (making the trained model more stable). The optimal model was then selected for model validation.

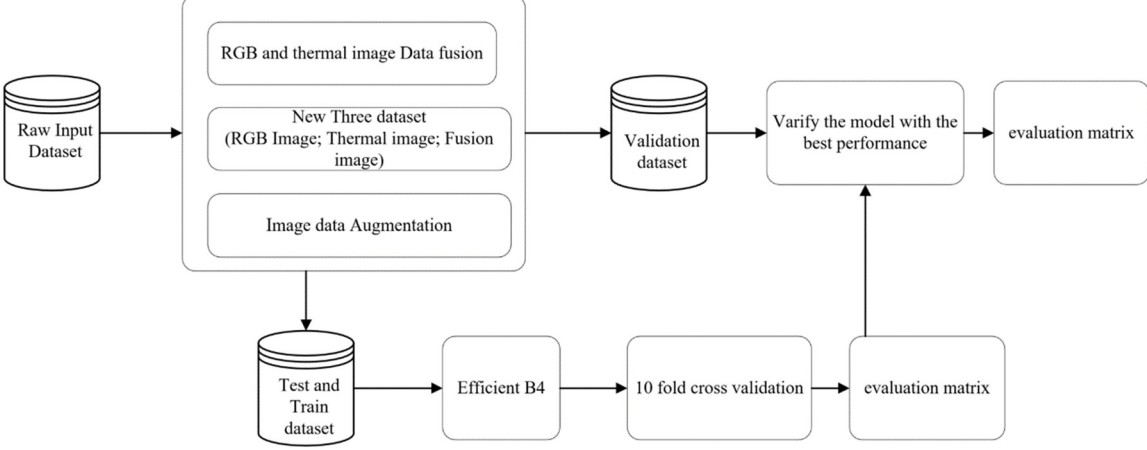

**Figure 1.** Work flow of proposed work.

### 2.2. Experimental Setup and Data Collection

All pavement data were acquired in Liverpool, UK (marked as red dots in Figure 2). Nine pavements were considered, namely (1) Transverse Cracks, (2) Longitudinal Cracks, (3) Alligator Cracks, (4) Joint or Patches, (5) Potholes, (6) Manholes, (7) Shadows, (8) Road Markings and (9) Oil Stains. The FLIR ONE camera connected to a cell phone was used for data capture wherein FLIR ONE's in-built application displays real-time thermal infrared images and saves both RGB and thermal images in one image by default. The images were then extracted through the MATLAB API interface [36] provided by FLIR. Since this experiment uses a passive heat source for the acquisition, the temperature data acquired is dependent on the collection time. In this case, the data was collected during the months of May and June, 2021 when the ambient temperature was in the range of 22 to 28 degrees Celsius. A total of nine categories of pavement features were acquired, 500 RGB images and 500 thermal images for each category. After completing the data collection, the image was classified and labeled manually.

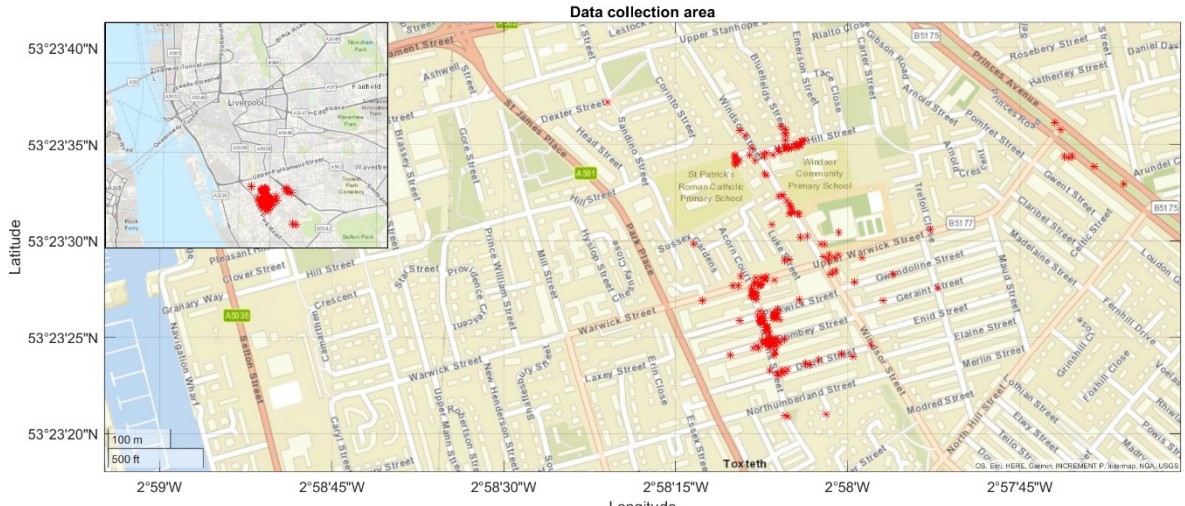

**Figure 2.** Data collection area.

### *2.3. Data Augumentation*

The captured images were affected by image acquisition system settings like non-uniform illumination, noise, etc., and the main one being the range of color bar for thermal images. As this dataset contains both RGB images and thermal images, different ways of data enhancement had to be performed.

For RGB images, illumination correction and addition of random noise were used for data augmentation as it was difficult to set up the light source that provides perfect illumination in the field. In the experiments performed in this paper, two-dimensional gamma functions were used to increase the variety of illumination scenes, which were further combined with the original image to simulate images with various non-uniform illumination effects [37]. The parameters of the sigma correction function were (15, 80, 150) and (70, 160, 250), respectively. Similar to the work presented by Song et al. [38], this experiment adds Gaussian noise to the original image to simulate noisy images with different signal-to-noise ratio (SNR) values (20 dB and 35 dB). Also, for thermal images, the adjusted color bar was randomly scaled down by 20% and expanded by 20% to expand the dataset.

In this experiment, there were 1500 RGB images and 1500 thermal images in each category after the expansion, totaling 13,500 images. The data classifier was trained using the original images and the expanded data in the dataset collected in 2.2. Only the collected data were used for evaluation to ensure the accuracy of the model and to avoid the existence of similar images in both train and test sets (the dataset was divided first and then the data was augmented. Figures 3–5 shows the results of data augmentation.

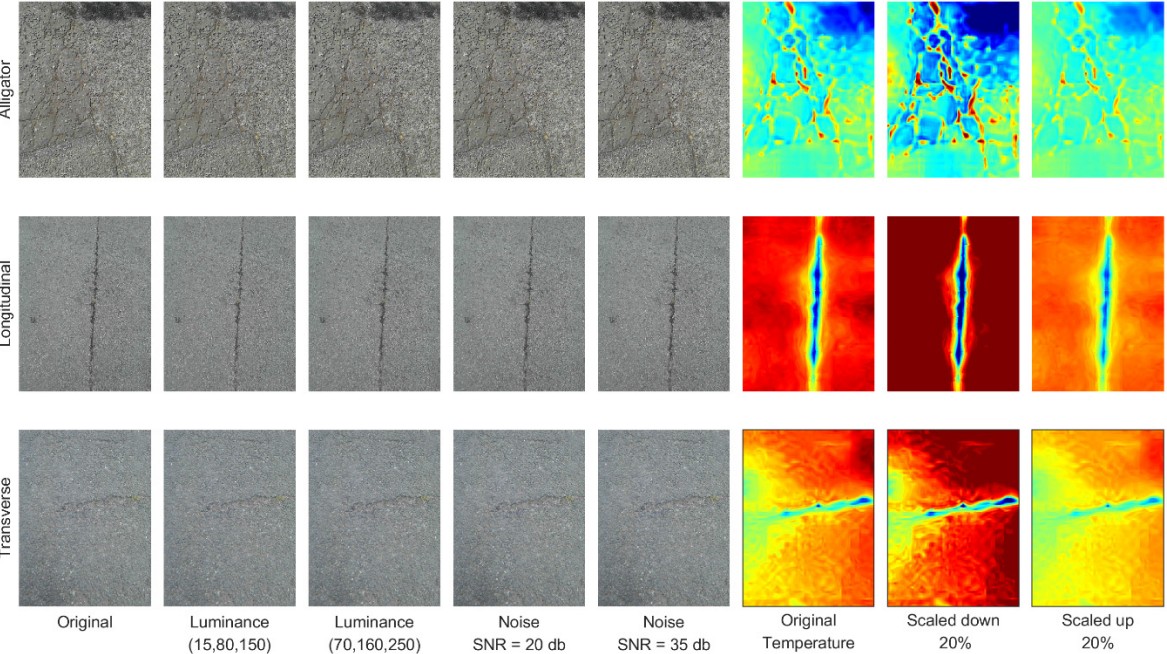

**Figure 3.** Data augmentation of alligator crack, longitudinal crack and transverse crack.

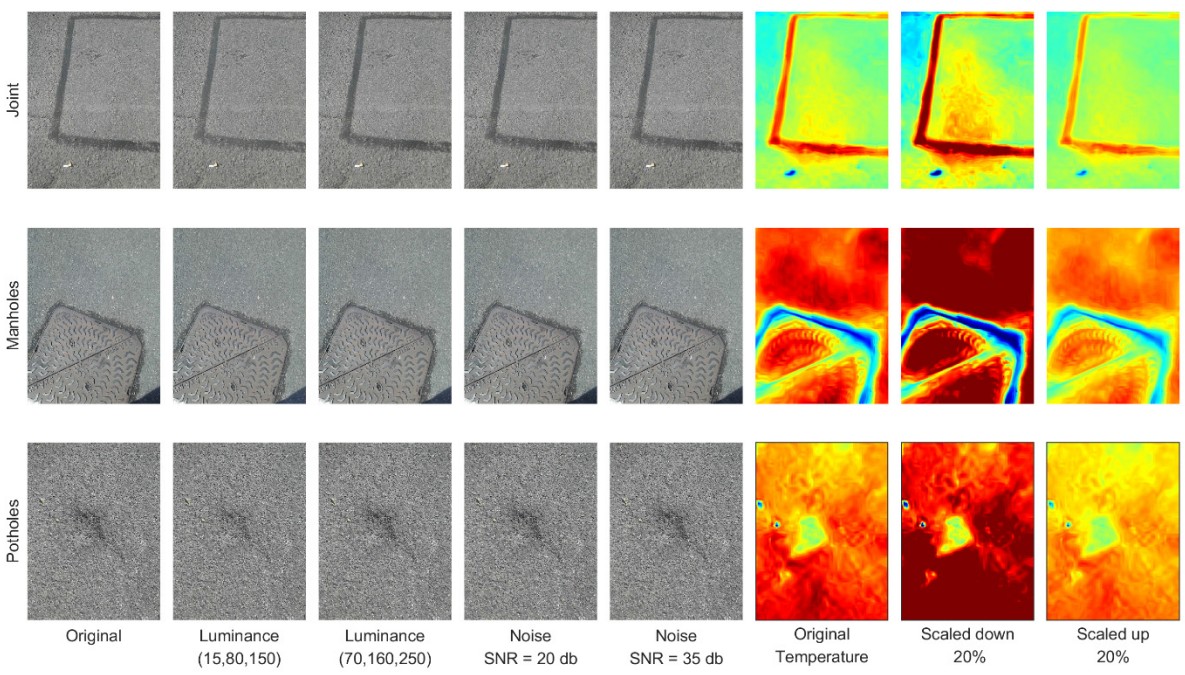

**Figure 4.** Data augmentation of joint, manholes and potholes.

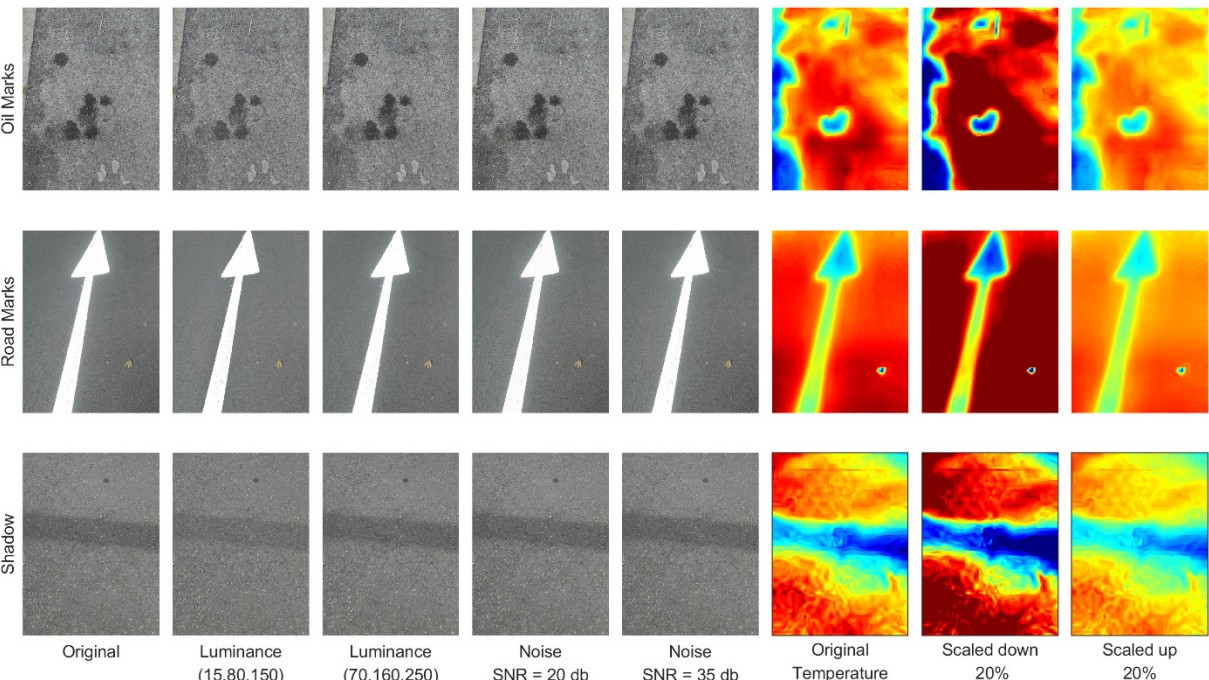

**Figure 5.** Data augmentation of oil marking, road marking and shadow.

### 2.4. Multi-Band Dynamic Imaging (MSX) and Its Adjustment

FLIR has the capability to add Multi-band Dynamic Imaging (MSX), which provides digital camera-level capture detail in thermal images [39]. MSX takes into account both the texture of the RGB image and the temperature of the thermal image, requiring adjustment of the ratio of the RGB and thermal images to adjust the visualization ratio of the image texture and the temperature field. Combining both RGB and thermal image information, MSX ensures easier target detection without compromising measurement data accuracy. Figure 6 shows the difference between MSX and RGB image and thermal image. Thus, as a

subsequent step of this paper, 50% transparency for the thermal image was used to fuse to form MSX image (performed under MATLAB script), and also augmented MSX images were formed based on the augmented image generated in Section 2.3.

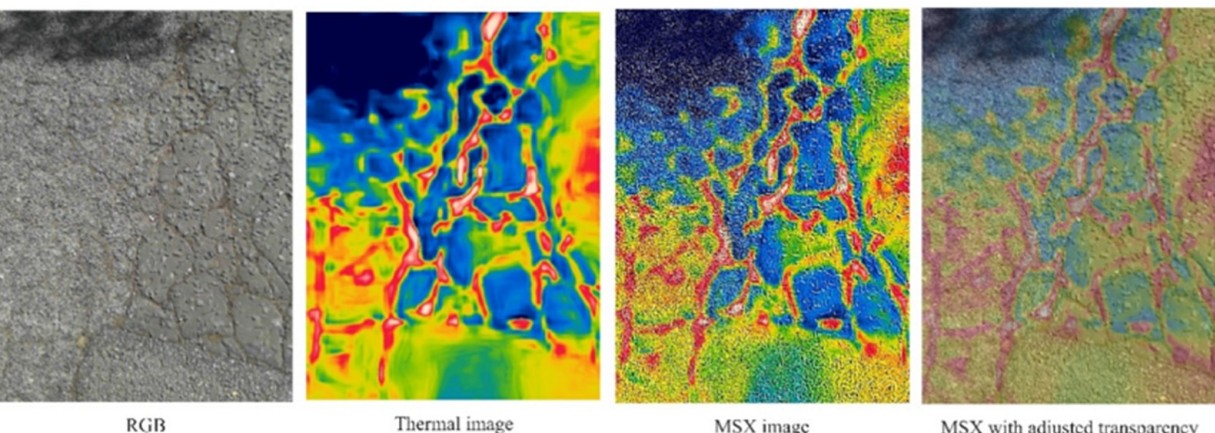

**Figure 6.** Samples of RGB image, thermal image, MSX image and MSX image with transparency.

### 2.5. Framework for Detection Network Based on EfficientNet

The most commonly used methods within deep learning for improving the model accuracy include expanding the network width, deepening the network depth, and increasing the resolution of input images. EfficientNet is a family of recurrent neural networks wherein the Efficient net models scale efficiently in terms of layer depth, layer width, input resolution, and a combination of all these factors [40]. Also, the EfficientNet models utilise a simple and efficient compound scaling method to scale the baseline ConvNet to any target, while maintaining efficiency. In general, the EfficientNet can be more accurate and efficient than existing CNNs such as AlexNet, ImageNet, GoogleNet and MobileNet [41,42]. Efficient Net-B0 was produced by Network Search [43] and versions from B0 to B7 continue to develop. The main building block of the network was that it consists of MBConv with the addition of compression and excitation optimization. Also, the backbone of EfficientNet contains 7 blocks, which in-turn have a different number of sub-blocks that increase with EfficientNetB0 to EfficientNetB7. The baseline model B0 of EfficientNet is shown in Figure 7. In this paper, we use EfficentNet B4 which contains 19M parameters as it accommodated our resources and usage.

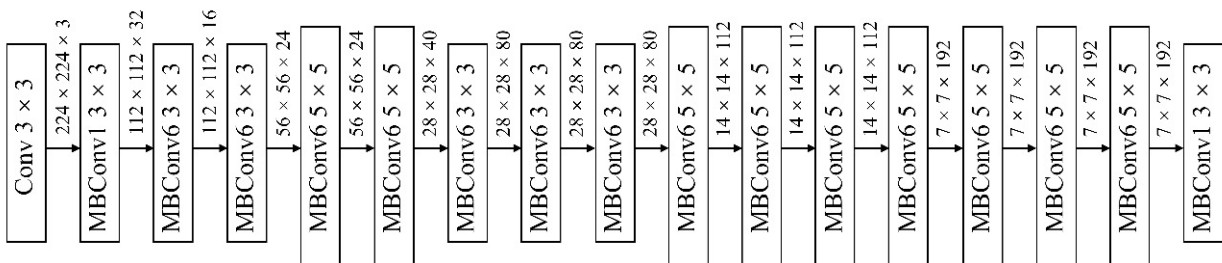

**Figure 7.** Baseline model EfficientNet B0.

The inverted bottleneck MBConv, the main building block for EfficientNet, consists of a layer that first extends and then compresses the channels, thus direct connections between bottlenecks can be used to connect much fewer channels than the extended layer. Compared to traditional convolutions layers, this architecture has deeply separable convolutions that reduce the computation by a factor of nearly k with conventional layers.

Also, compound scaling includes evenly scaling depth, width, and resolution (refer to Equations (1)–(4)). Atila et al. [44] describes the method for determining the scaling factor,

where $\alpha$, $\beta$ $\gamma$ $\alpha$, $\beta$, are constants determined by a grid search. Similarly, the user-defined factor, $\psi$, controls the available resources for model scaling, while $\alpha$, $\beta$, $\alpha$ $\gamma$, $\beta$ determine that the additional resources are allocated for network width, depth, and resolution, respectively. Starting from the baseline EfficientNet-B0, the compound scaling method scales.

EfficientNet-B0 in the following steps: assuming twice as many available resources, perform the grid search with $\psi$ = 1 and the optimal values of $\alpha$, $\beta$, $\gamma$. The $\alpha$, $\beta$, and $\gamma$ values are constants, and the baseline network can be scaled up to obtain EfficientB1-b7 with different $\psi$ values.

$$\text{depth} : d = \alpha^{\psi} \tag{1}$$

$$\text{width} : w = \beta^{\psi} \tag{2}$$

$$\text{resolution} : r = \gamma^{\psi} \tag{3}$$

$$\alpha \geq 1, \ \beta \geq 1, \ \gamma \geq 1 \tag{4}$$

## 3. Difference between Infrared Thermal Imaging and Optical Image Imaging in Complex Pavement Conditions

Conventionally, thermal imaging determines the temperature of an object by measuring the infrared radiation emitted from the surface of the target. Thermography measures temperature quickly enough to meet the purpose of real-time detection. Also, the Infrared ray (IR) is a band of invisible light found on the electromagnetic spectrum, with wavelengths ranging from 0.75 to 100 [45]. Thermal infrared cameras can detect infrared radiation from the short-wave to the long-wave region, wherein the infrared detector receives the infrared radiation energy distribution of the target, the sensor converts the radiation into an electrical signal, and the output gets processed to form an infrared thermal image. This thermal image corresponds to the heat distribution field on the surface of the object. This thermal imaging merits from having high portability, non-contact temperature measurement, absence of harmful radiation, real-time image acquisition capability and image independence from light intensity [46]. Also, with the relatively inexpensive cost of thermal infrared sensors makes them suitable in the field of identification where the accuracy of the temperature field is not as demanding.

The characteristics of thermal images and optical images are presented in Figure 8a–c. It is evident from Figure 8a that the crack region of the alligator crack has a higher temperature than the normal pavement, as the exposed asphalt can have higher heat absorption properties. Also, Longitudinal crack and transverse crack show a similar distribution of thermodynamic images with the exception of the direction of the crack. Also, the crack area appears to have a lower temperature due to the homogeneous pavement fracture into two parts with uneven thermal conductivity. Furthermore, the joint in Figure 8b has a higher temperature than the normal pavement area, which also applies to the repaired cracks due to the better heat absorption properties of the asphalt material. Manhole, on the other hand, showed a lower temperature in the manhole area than the normal pavement area due to the prevalence of separation between them. Additionally, the potholes area appears to have higher roughness and slower heat absorption properties resulting in a lower temperature profile. Similarly, in Figure 8c, the oiled areas showed higher heat absorption properties and as a result, displayed higher temperature profiles Also, the pavement markings (often as white or yellow markings) displayed lower temperature (than normal pavement) due to their low heat absorption performance. Finally, as the shadow blocks the sunlight, they displayed a lower temperature profile than the normal pavement.

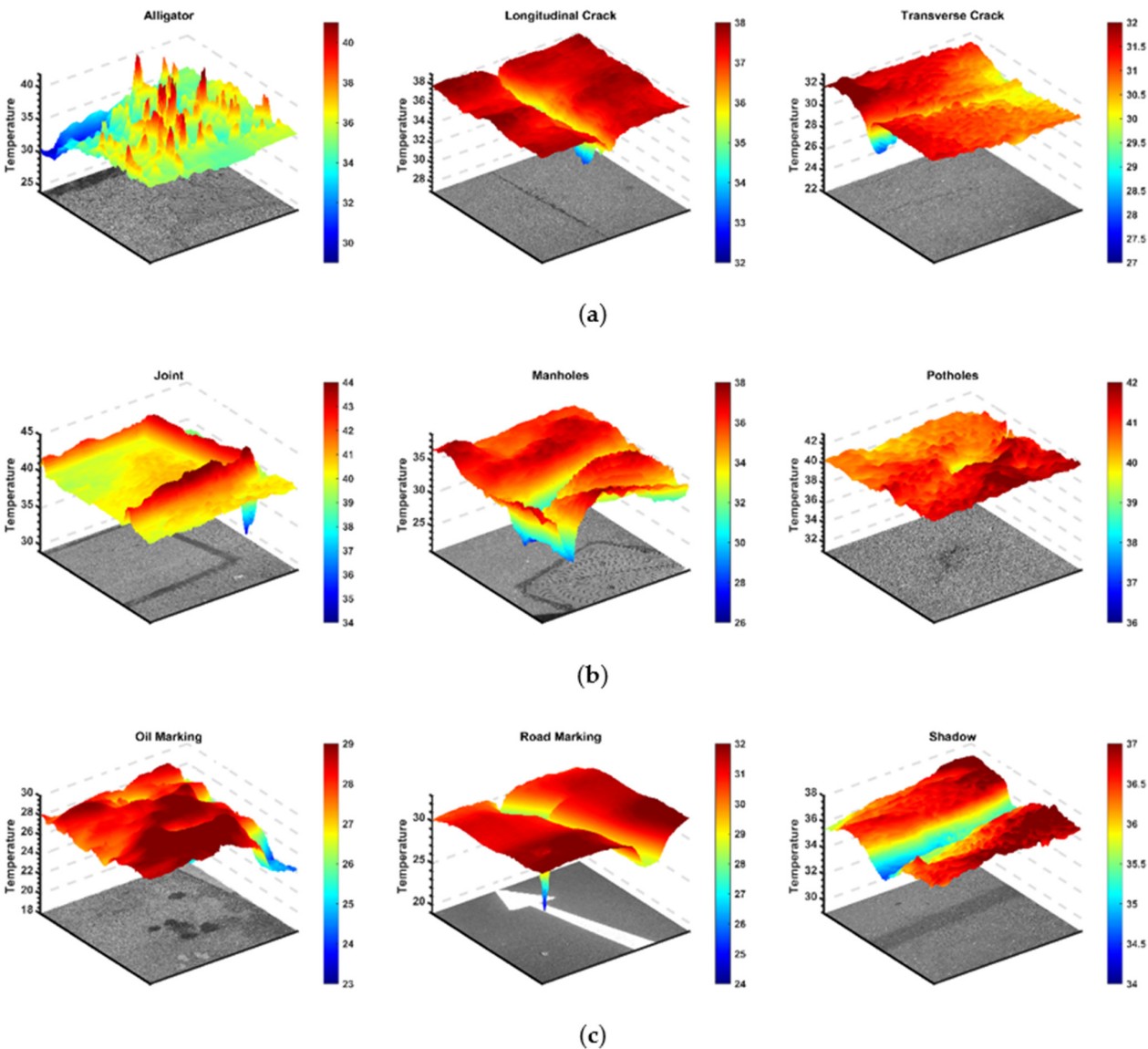

**Figure 8.** Optical and 3D thermal images of different pavement conditions. (**a**) 3D thermal image and RGB image of alligator crack, longitudinal crack and transverse crack. (**b**) 3D thermal image and RGB image of joint, manholes and potholes. (**c**) 3D thermal image and RGB image of oil marking, road marking and shadow.

Figure 9 presents the differential distribution statistical histogram of thermal image and RGB image respectively of Figure 8. In the RGB image, except for the grayscale of road marking which has a relatively large value, the other types have smaller grayscale values. The main reason for this being the crack recognition method based on histogram analysis tends to focus only on the grayscale differentiation of the road surface. Furthermore, the thermal image exhibits a higher temperature profile for exposed asphalt areas and oiled areas whereas lower measures for road marking, shadows, transverse cracks, and longitudinal cracks than normal pavement, thus gainfully complementing the single RGB image.

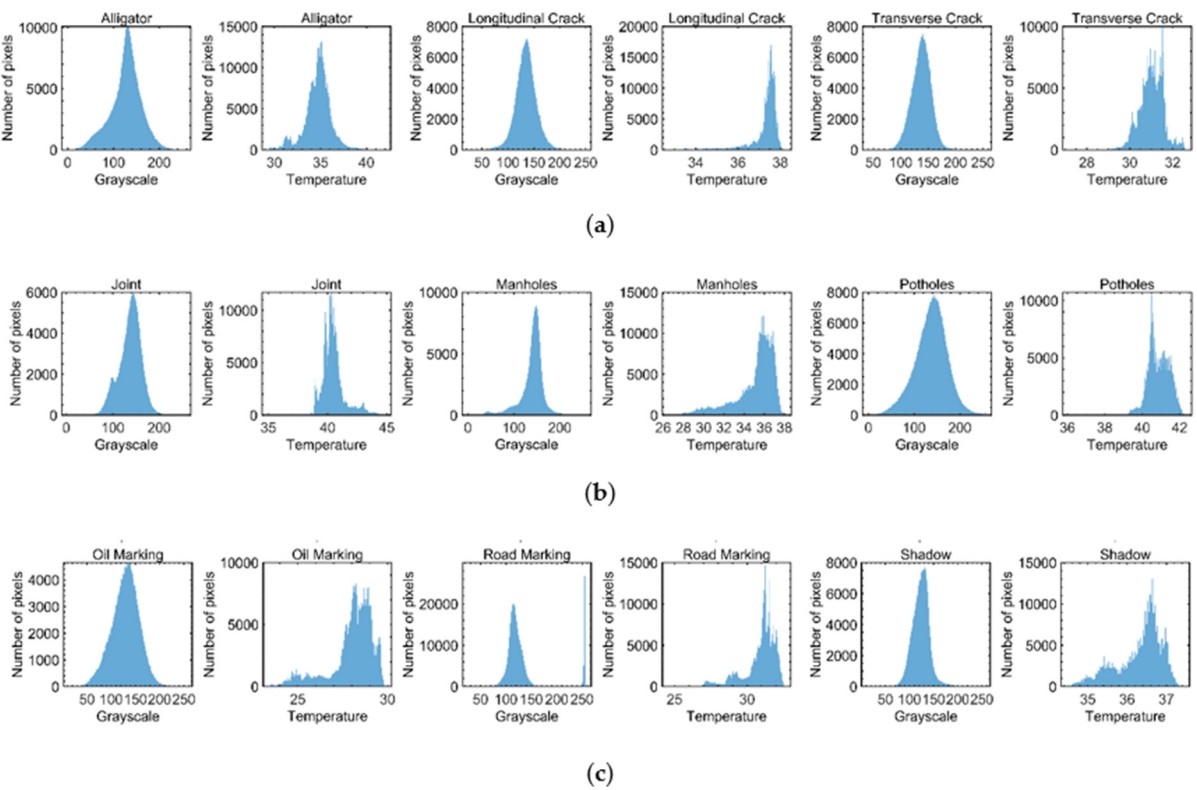

**Figure 9.** Numerical distribution of thermal image and RGB image. (**a**) Numerical distribution of thermal image and RGB image of alligator crack, longitudinal crack and transverse crack. (**b**) Numerical distribution of thermal image and RGB image of joint, manholes and potholes. (**c**) Numerical distribution of thermal image and RGB image of oil marking, road marking and shadow.

## 4. Results

### 4.1. Evaluation Metrics

Similar to works proposed by [6,41], the performance of our proposed models were evaluated using metrics like accuracy, precision, recall, and F1 score based on their respective true positive (*TP*), false positive (*FP*), true negative (*TN*) and false negative (*FN*) values. These metrics were computed as follows:

$$\text{Accuracy} = \frac{TP + TN}{TP + TN + FP + FN} \tag{5}$$

$$\text{Precision} = \frac{TP}{TP + FP} \tag{6}$$

$$\text{Recall} = \frac{TP}{TP + FN} \tag{7}$$

$$\text{F}_{1-\text{score}} = 2 \cdot \frac{\text{Precision} \cdot \text{Recall}}{\text{Precision} + \text{Recall}} \tag{8}$$

Having performed data preprocessing on MATLAB 2020a to form the dataset, python was used for deep learning operation and an Efficient module was executed using PyTorch framework on I9 10850K CPU and 2080ti GPU systems. The SGD algorithm was then used to update the network weight with momentum 0.7 and weight decay $1 \times 10^{-4}$ for all our experiments. The initial learning rate was set to 0.01 and adjusted further along during training using a multi-learning rate strategy, where the initial learning rate gets multiplied by 0.6 for each iteration (max_iter were set to 30 k). Furthermore, the parameters of the encoder EfficientNet were initialised using a transfer learning method on ImageNet after pre-training. Also, other parameters were initialized randomly. Finally, the batch size of the

dataset was set to 8, and to avoid overfitting, the data was scaled up by the steps described in Section 2.3.

The original and augmented datasets used in this study were randomly divided into training, validation, and test sets, accounting for 60%, 20%, and 20% spilt respectively. The training and validation datasets were used only for training and installation of the model, while the test set was used to evaluate the predictive performance of samples that were not previously seen by the model. Table 1 presents the original and augmented dataset size (inclusive of RGB, IR, and MSX images).

**Table 1.** Description of number of raw dataset and augumented dataset.

| Data Set | Total | Train (60%) | Validation (20%) | Test (20%) |
|---|---|---|---|---|
| Original dataset | 13,500 | 8100 | 2700 | 2700 |
| Augmented dataset | 18,945 | 11,367 | 3789 | 3789 |

### 4.2. Evaluation of Proposed RGB-Thermal Fusion-Based Image Detection Model

Although different versions of EfficientNet were explored under this research, EfficientNet B4 was chosen as the best encoder option for the main framework in this paper. For the model to converge quickly at the beginning, the dataset was then randomly disrupted before importing for calculation. This ensured that the input contains multiple types of data in a batch. As presented in Figure 10, the model was able to attain good accuracy after 3 epoch training. Furthermore, the algorithm was written to obtain results after 10-fold cross-validation before selecting the optimal one for detection. The results were then presented as an evaluation matrix (see Figure 11). The results showed a high average accuracy measure of 98.34% ranging from 96.91% to 99.52%; however, longitudinal and transverse cracks appear to have certain confusion as 11 images of longitudinal crack were misclassified as transverse crack. The reasons for these misclassifications could be that both longitudinal and transverse cracks exhibit similar texture and thermal characteristics with just the crack orientation seeming different and the angle of crack deflection seemingly difficult to quantify in the dataset. Figure 10 shows the loss and accuracy curves during training and validation.

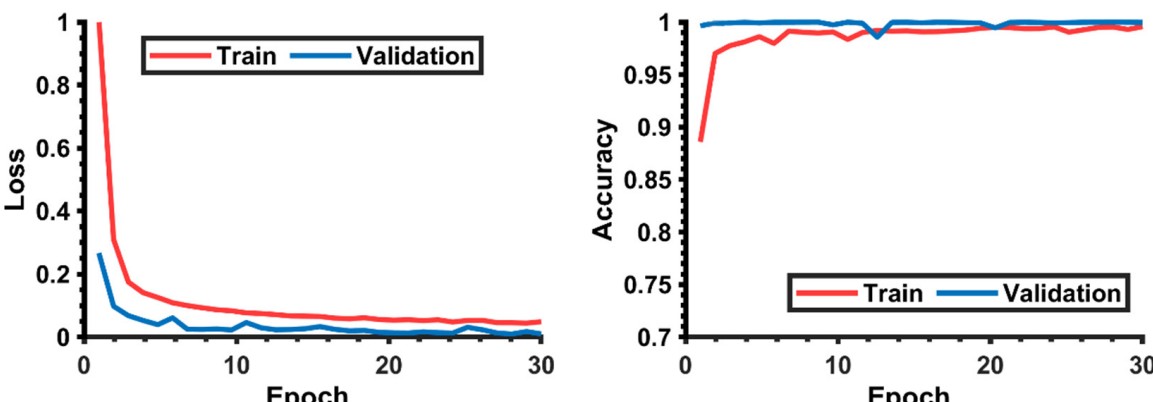

**Figure 10.** Training and validation curve of accuracy and loss.

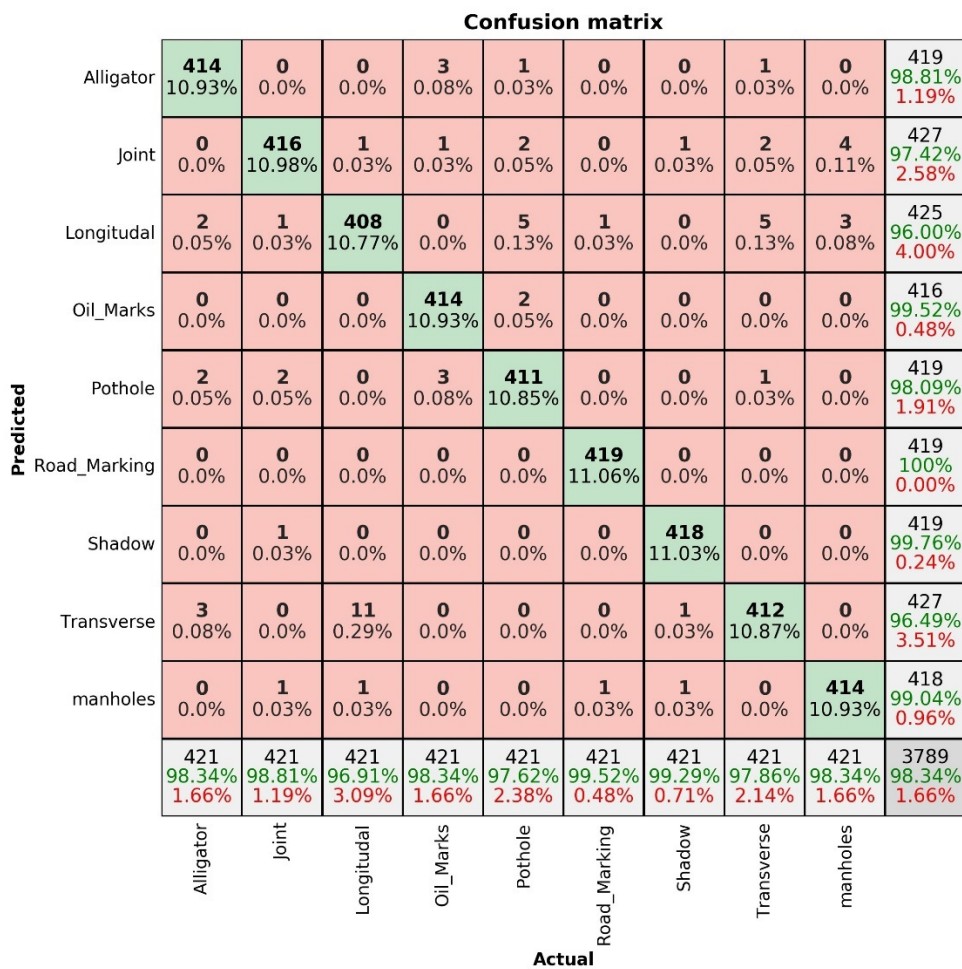

**Figure 11.** Evaluation matrix of proposed detection method.

### 4.3. Comparison of Different EfficientNet Versions in Pavement Crack Detection

To compare different network depths (EfficientNet B0 till B7) for the same type of network, experiments were performed using only RGB-thermal fusion-based images and the results are presented in Table 2. This proves that the performance of higher versions of the EfficientNet model was reduced due to the increased number of parameters and model complexity, thus making the search process as finding a balance between data samples and network parameters.

**Table 2.** Results for pavement defect detection on original dataset.

| Model | Accuracy | Precision | Recall | F1-Score |
|---|---|---|---|---|
| EffecientNet B0 | 95.36% | 95.97% | 95.24% | 95.60% |
| EffecientNet B1 | 96.55% | 97.11% | 95.62% | 96.36% |
| EffecientNet B2 | 95.69% | 96.04% | 96.38% | 96.21% |
| EffecientNet B3 | 97.31% | 96.91% | 95.89% | 96.40% |
| EffecientNet B4 | 98.65% | 98.34% | 97.14% | 97.74% |
| EffecientNet B5 | 98.92% | 98.22% | 98.56% | 98.39% |
| EffecientNet B6 | 97.92% | 97.94% | 97.84% | 97.89% |
| EffecientNet B7 | 98.15% | 97.22% | 98.32% | 97.77% |

As shown in Table 2, the average accuracy values using the original dataset for all models were high, with EfficientNet achieving the best results in B5. For the network comparison, as EfficientB0 to B7, the number of parameters to be trained becomes larger, thus requiring a longer time to train. Also, evident from Table 2, the EfficientNet B5 model

provides the best average sensitivity values in comparison to other classes (increase true positive classification). Also, the B5 model demonstrated to exhibit higher precision, recall, and F1-score. Extending this experimentation with the augmented dataset evaluation (presented in Table 3) showed that B4 Model achieved similar results to B5 with all evaluation parameters being higher than 98%. Despite similar results, B4 had to be considered as a better recognition network as it can train faster with fewer training parameters than B5. Also, the precision, recall, and F1 values of the model with the augmented data can be deemed more stable with strengthened model resistance to interference.

**Table 3.** Results for pavement defect detection on augmented dataset.

| Model | Accuracy | Precision | Recall | F1 |
|---|---|---|---|---|
| EffecientNet B0 | 96.21% | 95.68% | 95.77% | 95.72% |
| EffecientNet B1 | 96.57% | 96.59% | 96.57% | 96.58% |
| EffecientNet B2 | 96.57% | 96.59% | 96.57% | 96.58% |
| EffecientNet B3 | 97.23% | 97.12% | 97.14% | 97.13% |
| EffecientNet B4 | 98.34% | 98.35% | 98.34% | 98.34% |
| EffecientNet B5 | 98.33% | 98.23% | 98.12% | 98.17% |
| EffecientNet B6 | 98.11% | 98.01% | 98.11% | 98.16% |
| EffecientNet B7 | 98.02% | 97.21% | 97.77% | 97.49% |

## 5. Discussion

Each of the four types of data (indicated in Figure 6) was used as input data They were then put through the optimal model identified in Section 4.3, and the results are presented in Table 4 evidently, the RGB input had better prediction results than IR with higher accuracy, precision, recall, and F1-score. Consequently, MSX images enhanced the evaluation results of recognition rate in comparison to RGB images. Finally, the fused image (adjusted the transparency to show both texture and thermal properties) input exhibited significantly higher and better detection results than any other input images types (RGB, IR, and MSX).

**Table 4.** Comparison of different input data in pavement crack detection.

| Image Type | Accuracy | Precision | Recall | F1 |
|---|---|---|---|---|
| RGB | 96.57% | 96.59% | 96.57% | 96.57% |
| IR | 93.83% | 93.95% | 93.83% | 93.84% |
| MSX | 96.83% | 96.92% | 96.83% | 96.83% |
| Fused image | 98.34% | 98.35% | 98.34% | 98.34% |

Analysing the same by pavement feature category, Table 5 presents the comparison of the model's prediction performance of the different input types for each class. Also, Figure 12 shows a detailed visual comparison of the model's prediction performance for each input image type. Overall, the results prove that the fused images as input achieve best results than other input image types with reliable predictions for alligator crack, joint, longitudinal crack, pothole, and transverse crack categories. However, the prediction performance for road marking, shadow, and manholes were slightly weaker than RGB images. This also proves that the fused images can indeed provide more information to differentiate pavement features better with more stable recognition performance for complex conditions too.

**Table 5.** Comparison of performance of different input data for each class.

|                  | RGB    | IR     | MSX    | FUSION |
|------------------|--------|--------|--------|--------|
| Alligator crack  | 96.83% | 95.24% | 96.83% | 98.34% |
| Joint            | 93.65% | 90.48% | 90.48% | 98.81% |
| Longitudinal     | 93.65% | 92.06% | 87.30% | 96.91% |
| Oil Marking      | 98.41% | 90.48% | 98.41% | 98.34% |
| Pothole          | 95.24% | 87.30% | 95.24% | 97.62% |
| Road Marking     | 100%   | 96.83% | 100%   | 99.52% |
| Shadow           | 100%   | 98.41% | 98.41% | 99.29% |
| Transverse       | 90.48% | 95.24% | 96.93% | 97.86% |
| Manholes         | 100%   | 98.41% | 93.65% | 98.34% |
| Average          | 96.47% | 93.83% | 95.24% | 98.34% |

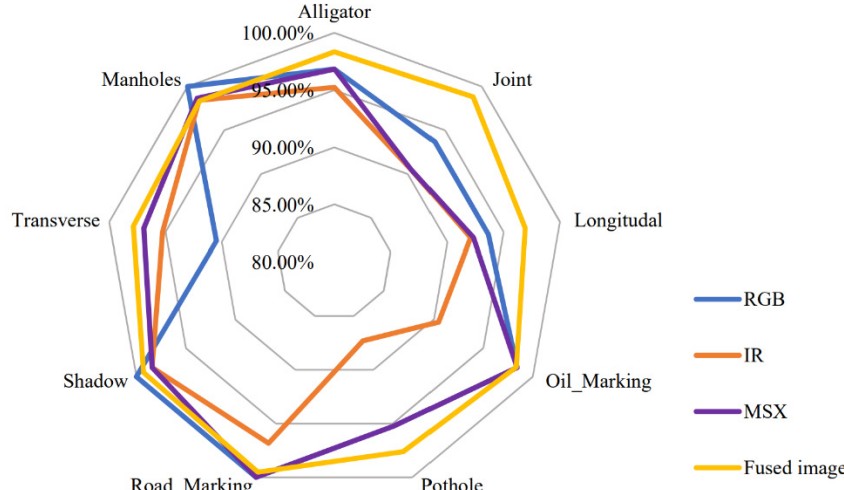

**Figure 12.** Comparison of performance for each class with different input data.

*Comparison of Thermal RGB Fusion with Other Pavement Crack Detection Methods*

Comparing the obtained results with other mainstream pavement defect detection systems (employing sensors) like 3D camera, laser camera, 3D laser scanning, acceleration sensors shows that the method proposed in this paper achieves similar accuracy levels (Table 6). Additionally, unlike other techniques, this proposed method accommodates complex pavement scenarios too. Also, the post-processing process of the data was more straightforward with collected images readily recognisable without pre-processing in comparison with point cloud images or vibration data.

**Table 6.** Comparison of other mainstream learning-based detection method with proposed method.

| Methods             | Data Type                        | Accuracy (%)    |
|---------------------|----------------------------------|-----------------|
| Zhang et al. [47]   | RGB image (Kinect-Based)         | 89.09           |
| Zhou and Song [22]  | laser-scanned range images       | 99.6 (average)  |
| Zhong et al. [9]    | 3D laser scanning                | Over 98         |
| Chen et al. [6]     | vibration signal                 | 97.2            |
| Our method          | RGB image with thermal information | 98.34%        |

## 6. Conclusions

Deep learning methods are increasingly being to pavement detections; however, the prediction accuracy of previous image-based detection methods tends to be compromised due to its inability to accommodate complex and large pavement features/scenes. This paper proposes an EfficientNet B4-based detection model for pavement damage recognition using a thermal image-assisted fusion of RGB images. The effect of model, data

enhancement, and data source on the recognition performance is illustrated in detail by comparing the results.

This paper has proposed an efficient net deep learning architecture to classify nine classes of pavement features from the collected pavement dataset. Considering the average accuracy and average precision metrics of the original and augmented datasets, the EfficientNetB5 model achieved 98.92% accuracy and 98.22% precision on the original dataset, while the EfficientNetB4 model achieved 98.34% accuracy and 98.35% precision on the augmented dataset. Also combining the performance of other networks on the augmented data increased the recognition stability of the model. Since EfficientNetB4 has fewer training parameters, efficientB4 has the best performance on EfficientNet.

The IR image enhanced RGB image provides more image information compared to RGB image, IR image, or MSX image. Better Accuracy, Precision, Recall, and F1Score were achieved using the fused images.

The proposed method is based on thermal and RGB fused images and achieves the same accuracy as laser scanner when using sensor image overlay. Date from multiple sensors can provide more valid information for complex scenarios. Also, the proposed method is completely based on images; the processing is fast. But layer superposition of data from different sources will undoubtedly lose some of the original information, and using fusion model to handle different source sensors data to achieve better detection accuracy.

**Author Contributions:** Conceptualization, C.C. and H.S.; Software and coding, C.C. and S.C.; writing original draft, C.C.; Data collection. C.C.; Data pre-processing, C.C. and S.C.; Draft correction & review: C.C., Y.H. and H.S.; Visualization, C.C.; Supervision, H.S.; Final report approval, H.S. All authors have read and agreed to the published version of the manuscript.

**Funding:** This research received no external funding.

**Institutional Review Board Statement:** Not applicable.

**Informed Consent Statement:** Not applicable.

**Data Availability Statement:** Data available on request due to restrictions.

**Conflicts of Interest:** The authors declare that there were no known competing financial interest or personal relationships that could have influenced the work in this paper.

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
