# Peer review of "Deep Learning-Based Thermal Image Analysis for Pavement Defect Detection and Classification Considering Complex Pavement Conditions"

_remotesensing, doi:10.3390/rs14010106_

Round 1

Reviewer 1 Report

The automatic detection and identification of pavement distress is an important issue in evaluating pavement condition for maintenance and rehabilitation purposes. The manuscript aims to present an innovative method for pavement defects detection based on thermal image and deep learning. It is a high research topic on Artificial Intelligence and, in particular, Machine Learning algorithms. I believe that image analysis has great potential for practical implementation in pavement engineering, but it is a very complex methodology that demands good accuracy.

The paper describes the method and its application to several types of distresses, validating the methodology and demonstrating that it is more accurate than other methods.

The manuscript presents good research and is well written and organized.

The paper is well written and organized. Some comments are addressed to the authors to clarify and improve the quality and scientific merit of the paper.

1 - It is recommended to check the paper for errors correction. Example: always leave a blank space between words (…”segmentation.Especially”…), there are several examples of this error; captions of Figure 3 (“augumentation” and alligator crack); some references in the text are underlined; Equation 1 is, in fact, three equations, please consider to split them. Line 310: true positive; the size of the font in the tables is excessive; caption of Figure 9 (matrix); the title of sub-section 4.3 (…”versions in pavement…)

2 - I recommend splitting Figure 3 into three figures to better read and understand.

3 – The introduction contains the literature review. My suggestion is to include more international references to demonstrate that important advances are also being achieved on the paper topic in other regions, such as the USA and Europe.

4 – The conclusions should also include the authors' opinion regarding some advantages of the proposed method in comparison with other equivalent methods and future works for research and development.

Author Response

The responses were attached.

Reviewer 2 Report

Deep learning-based thermal image analysis for pavement defect detection and classification considering complex pavement conditions

A brief summary

The paper proposes a thermal-RGB fusion image-based pavement damage detection model, wherein the fused RGB-thermal image formed through multi-source sensor information, to achieve fast and accurate defect detection including complex pavement conditions. The proposed method uses pre-trained EfficientNet B4 as the backbone architecture and generates an argument dataset to achieve high pavement damage detection accuracy.

Comments

Strengths of the paper:

  1. Professionally laid out paper – both academic and practical, comprehensive, clear and solid in terms of descriptive content and proposed considerations / applications.
  2. Extensive review references (38) on the considered topic and a clear review of published research works presented.
  3. Well thought out methodology by using different approaches and methods.
  4. The proposed method seems to be innovative and contains well-known hints of originality.
  5. The authors have been found promising results and have been evaluated them through the parameters usually used in automatic distress pavement detection: Precision 98.35%, Recall 98.34% and F1-score 98.34%..

Weakness of the paper:

  • The authors should inform the paper's readers about the devices used to capture the RGB and thermal images.
  • The paper text should be revised paying attention because there are some typo and style errors like missed spaces between words e parenthesis, missing dots at the end of the phrases, words underlined when not required, ecc.
  • The style used to recall references in the paper seem to be different to that indicated by the Journal in the guide for the authors.
  • The authors should mention how the ground truth data have been obtained. Which difficulties are there which are making pavement damage detection difficult?
  • References could be enriched by adding in the literature also some recent researches on this topic. I suggest considering these papers:
    1. Fan, Z.; Li, C.; Chen, Y.; Wei, J.; Loprencipe, G.; Chen, X.; Di Mascio, P. Automatic Crack Detection on Road Pavements Using Encoder-Decoder Architecture. Materials 2020, 13, 2960.
    2. Wang, W.J.; Su, C.; Semi-supervised semantic segmentation network for surface crack detection, Automation in Construction 128 (2021).
    3. Ali, R.; Chuah, J.H.; Talip, M.S.A.; Mokhtar, N. Shoaib, M.A. Automatic pixel-level crack segmentation in images using fully convolutional neural network based on residual blocks and pixel local weights, Engineering Applications of Artificial Intelligence 104 (2021).
  • … no more weaknesses!

The overall merit of presented research works and findings is high and definitely worth publishing after incorporation the above minor suggestions.

Author Response

The responses were attached.
